# The characteristics and evolution of pulmonary fibrosis in COVID-19 patients as assessed by AI-assisted chest HRCT

**Jia-Ni Zou[1]◉, Liu Sun[2]◉, Bin-Ru Wang[3]◉, You Zou[4]◉, Shan Xu[4], Yong-Jun Ding[4], Li-Jun Shen[4], Wen-Cai Huang[1], Xiao-Jing Jiang[5], Shi-Ming Chen◉[4,6]***

**1** Department of Radiology, General Hospital of Central Theater Command of the PLA, Wuhan, Hubei, P. R. China, **2** Department of Otolaryngology-Head and Neck Surgery, General Hospital of Central Theater Command of the PLA, Wuhan, Hubei, P. R. China, **3** Department of Otolaryngology-Head and Neck Surgery, Tongren Hospital of Wuhan University (Wuhan Third Hospital), Wuhan, Hubei, P. R. China, **4** Department of Otolaryngology-Head and Neck Surgery, Renmin Hospital of Wuhan University, Wuhan, Hubei, P. R. China, **5** Department of Infectious Diseases, General Hospital of Central Theater Command of the PLA, Wuhan, Hubei, P. R. China, **6** Institute of Otolaryngology-Head and Neck Surgery, Renmin Hospital of Wuhan University, Wuhan, Hubei, P. R. China

◉ These authors contributed equally to this work.
* shimingchen0468@163.com

**Data Availability Statement:** All relevant data are within the paper and its Supporting Information files.

## Abstract

The characteristics and evolution of pulmonary fibrosis in patients with coronavirus disease 2019 (COVID-19) have not been adequately studied. AI-assisted chest high-resolution computed tomography (HRCT) was used to investigate the proportion of COVID-19 patients with pulmonary fibrosis, the relationship between the degree of fibrosis and the clinical classification of COVID-19, the characteristics of and risk factors for pulmonary fibrosis, and the evolution of pulmonary fibrosis after discharge. The incidence of pulmonary fibrosis in patients with severe or critical COVID-19 was significantly higher than that in patients with moderate COVID-19. There were significant differences in the degree of pulmonary inflammation and the extent of the affected area among patients with mild, moderate and severe pulmonary fibrosis. The IL-6 level in the acute stage and albumin level were independent risk factors for pulmonary fibrosis. Ground-glass opacities, linear opacities, interlobular septal thickening, reticulation, honeycombing, bronchiectasis and the extent of the affected area were significantly improved 30, 60 and 90 days after discharge compared with at discharge. The more severe the clinical classification of COVID-19, the more severe the residual pulmonary fibrosis was; however, in most patients, pulmonary fibrosis was improved or even resolved within 90 days after discharge.

## Introduction

Pulmonary fibrosis can occur as a serious complication of viral pneumonia, which often leads to dyspnea and impaired lung function. It significantly affects quality of life and is associated with increased mortality in severe cases [1, 2]. Patients with confirmed severe acute respiratory syndrome coronavirus (SARS-CoV) or Middle East respiratory syndrome coronavirus

**Funding:** This study was supported by grant 81770981 and 82002863 from the National Natural Science Foundation of China and grant [2018]116 from Wuhan Municipality Young and Middle-aged Medical Talent Cultivation Program.

**Competing interests:** No authors have competing interests

(MERS-CoV) infections were found to have different degrees of pulmonary fibrosis after hospital discharge, and some still had residual pulmonary fibrosis and impaired lung function two years later. In addition, wheezing and dyspnea have also been reported in critically ill patients [3–5].

Severe acute respiratory syndrome coronavirus 2 (SARS-CoV-2) is a novel Betacoronavirus that is responsible for an outbreak of acute respiratory illness known as coronavirus disease 2019 (COVID-19). SARS-CoV-2 shares 85% of its genome with the bat coronavirus bat-SL-CoVZC45 [6]. However, there are still some considerable differences between SARS-CoV-2 and SARS-CoV or MERS-CoV. Whether COVID-19 can trigger irreversible pulmonary fibrosis deserves more investigation. George reported that COVID-19 was associated with extensive respiratory deterioration, especially acute respiratory distress syndrome (ARDS), which suggested that there could be substantial fibrotic consequences of infection with SARS-CoV-2 [7]. Moreover, it has also been shown that the pathological manifestations of COVID-19 strongly resemble those of SARS and MERS [8], with pulmonary carnification and pulmonary fibrosis in the late stages.

Chest X-rays and high-resolution computed tomography (HRCT) of the chest play important auxiliary roles in the diagnosis and management of patients with suspected cases of COVID-19 [9, 10]. The newly applied artificial intelligence (AI)-assisted pneumonia diagnosis system has been described as an objective tool that can be used to qualitatively and quantitatively assess the progression of pulmonary inflammation [11]. At present, although COVID-19 has been classified as a global epidemic for months, the risk factors for and severity and evolution of pulmonary fibrosis have not yet been reported. In this study, this new technology was applied to investigate the pulmonary imaging characteristics and related risk factors in COVID-19 patients at the time of hospital discharge, as well as the evolution of pulmonary fibrosis 30, 60 and 90 days after discharge, with the aim of providing an important basis for the clinical diagnosis, treatment and prognostic prediction of COVID-19-related pulmonary fibrosis.

## Materials and methods

### Study subjects

All 284 patients who had confirmed cases of COVID-19 and achieved a clinical cure from February 1 to March 31, 2020, at the Central Theater General Hospital of the Chinese People's Liberation Army were recruited. Their clinical characteristics and chest HRCT data were collected; follow-up studies on the evolution of pulmonary fibrosis were conducted with patients who returned to the hospital for chest HRCT reexaminations 30 days, 60 days and 90 days after hospital discharge.

The clinical diagnosis, treatment, clinical classification and discharge criteria for all patients were based on the *Diagnosis and Treatment of COVID-19 (trial version 7)* published by the National Health Commission of China [12]. The inclusion criteria were as follows: primary infection with SARS-CoV-2 confirmed by a positive upper respiratory swab RT-PCR. The exclusion criteria were other viral respiratory infections, coexisting connective tissue disease, and a history of lung disease. The experimental procedures used in this study were approved by the ethics committee of Renmin Hospital of Wuhan University and Central Theater General Hospital of the Chinese People's Liberation Army (WDRY2020-K110), [2020]030–1). The collection and use of relevant case data were performed with adequate protection of the patients' privacy and met the ethics requirements. The need to obtain the informed consent of the patients was waived. All data were anonymized. The patients' medical records were accessed from February 1 to June 8, 2020.

## Chest HRCT examinations

In accordance with the COVID-19 Close Contacts Management Guidelines published by the National Health Commission of China [13], all patients underwent a chest HRCT examination in a designated room in which the environment and equipment were completely sterilized, and the scanning technicians were all wearing the appropriate level of personal protective equipment. The patients had to wear masks and were examined in the supine position after receiving instructions about breathing during the scan. A Sino-vision 64s CT spiral scanner (SINO VISION, Beijing) was used, and the scan covered the area from the apex of the lung to the costophrenic angle. The scanning parameters were as follows: tube voltage of 120 kV, adjusted tube current that ensured that the CTDIvol value was 7 mGy, scanning layer thickness and layer spacing of 0.5–2 mm, spiral pitch of 1.3, and pedal scan direction.

## Chest HRCT image analysis

Image analysis was performed independently by 2 senior diagnostic radiologists in a double-blinded fashion. When their opinions differed, the chief diagnostic chest imaging physician was asked to lead a discussion, during which a final agreement was reached. The chest CT image analysis included the distribution of the lesions, the location of the lesions, the number of lobes affected, the characteristics of the lesions and external involvement. For each patient, the CT presentation was described according to the following parameters.

**Lesion degree.**   The types of lesions were as follows: ground-glass opacity, linear opacity, interlobular septal thickening, reticulation, honeycombing or bronchiectasis. To describe the extent of the lesions, the lungs were divided into approximately 20 equal parts according to the distribution of lung segments (2 parts in the posterior apical segment of the left upper lobe, 2 parts in the anterior and basal segments of the lower lobe, and 1 part each for the upper and lower lingual segments of the left lung) [14].

**Quantitative scoring of pulmonary fibrosis.**   The degree of pulmonary fibrosis was evaluated using the CT scoring method proposed by Camiciottoli [15]: chest HRCT images (ground-glass opacity, linear opacity, interlobular septal thickening, reticulation, honeycombing or bronchiectasis) were independently scored by two people; the intraclass correlation coefficient (ICC) was approximately 0.99, which indicated that this method was reliable. The mean was taken as the final score when the scores were inconsistent.

The scoring method had two parts, one for the lesion type and the other for the extend of the lesions. The maximum score was 30. The types of lesions were ground-glass opacities, linear opacities, interlobular septal thickening, reticulation, honeycombing and bronchiectasis, which were scored as 1, 2, 3, 4 and 5, respectively. The extent of each type of lesion was scored based on whether that lesion type was identified in 1 ~ 3, 4 ~ 9 or more than 9 pulmonary segments, which were scored as 1, 2 and 3, respectively. For example, if there were ground-glass opacities in 1 to 3 lung segments, the pulmonary fibrosis score was 1+1 = 2. The total quantitative pulmonary fibrosis score was equal to the score for all types of lesions + the extent score for each type of lesion; the total score ranged from 0 to 30. Pulmonary fibrosis was classified into three groups based on the total score as follows: mild (0–10), moderate (11–20), and severe (21–30).

**AI inflammation score.**   The relative quantification was performed with the "Artificial Intelligence (AI)-assisted Pneumonia Diagnosis System" software developed by Hangzhou Etu Medical Technology Co. (https://www.yitutech.com). This system combines convolutional neural networks with the threshold method to dissect the left and right lungs and detect the areas of inflammation. It then calculates the pulmonary inflammatory volume (PIV), whole lung volume (WLV), percentage of diseased lung (PIV/WLV), and quantitative parameters such as the involved lung lobes and involved lung segments.

### Risk factors for pulmonary fibrosis

Patient clinical data were collected from the electronic medical records system. The general clinical data included sex, age, and main clinical symptoms. The laboratory findings are listed in Table 4. All laboratory results were measured within 48 hours of admission.

### Statistical analysis

Categorical variables are described by frequencies and percentages, and normally distributed data are presented as the means ± standard deviations (X±SD). Skewed data are presented as the medians and interquartile ranges. Comparisons between groups of measures that conformed to a normal distribution were performed using t tests, and comparisons between groups with skewed data distributions were performed with Wilcoxon's test or the Mann-Whitney U test. Categorical data were analyzed using the $\chi 2$ test and Fisher's exact test, and correlations were analyzed using Spearman's rank correlation analyses. Statistical analyses were performed with SPSS (version 26.0) software. Statistical significance was defined by a two-sided $P < 0.05$.

## Results

### The proportion of COVID-19 patients with pulmonary fibrosis

A total of 284 COVID-19 patients who achieved a clinical cure were enrolled in this study, of whom 239 (84.15%) had pulmonary fibrosis and 45 (15.85%) did not have fibrosis. Pulmonary fibrosis occurred in 169 patients with moderate COVID-19 (78.9%) and in all patients with severe or critical COVID-19 (100%). The incidence of pulmonary fibrosis in patients with moderate COVID-19 was significantly lower than that in patients with severe or critical COVID-19 (Table 1, P<0.01).

### The relationship between the degree of pulmonary fibrosis and the clinical classification of COVID-19

Among the 239 COVID-19 patients with pulmonary fibrosis, 169 patients with moderate COVID-19 had mild pulmonary fibrosis (101 patients, 59.76%) or moderate pulmonary fibrosis (68 patients, 40.24%). Fifty-seven patients with severe COVID-19 had mild pulmonary fibrosis (10 patients, 17.54%), moderate pulmonary fibrosis (25 patients, 43.86%) or severe pulmonary fibrosis (22 patients, 38.60%). Thirteen patients with critical COVID-19 had mild pulmonary fibrosis (3 patients, 23.08%), moderate pulmonary fibrosis (3 patients, 23.08%) or severe pulmonary fibrosis (7 patients, 53.85%) (Table 1). These results indicated that patients with moderate COVID-19 mainly developed mild-to-moderate pulmonary fibrosis, while patients with critical COVID-19 generally developed severe pulmonary fibrosis (P<0.01).

### Pulmonary fibrosis quantitative scores

Among the 239 COVID-19 patients with pulmonary fibrosis, those with mild cases mainly had ground-glass opacities and linear opacities; those with moderate cases mainly had ground-glass opacities, linear opacities and interlobular septal thickening; and those with severe cases mainly had ground-glass opacities, linear opacities, interlobular septal thickening, reticulation, honeycombing or bronchiectasis. There were significant differences in the lung segments affected by linear opacities or interlobular septal thickening among patients with mild, moderate, and severe cases (Table 2, P<0.01).

**Table 1. The proportion of COVID-19 patients with pulmonary fibrosis at discharge and its relationship to the clinical classification.**

| Pulmonary fibrosis | | Groups | | | Total | $\chi^2$ value | P value |
|---|---|---|---|---|---|---|---|
| | | Moderate COVID-19 | Severe COVID-19 | Critical COVID-19 | | | |
| Yes | Mild[a] | 101 | 10 | 3 | 114 | | |
| | Moderate[b] | 68 | 25 | 3 | 96 | | |
| | Severe[c] | 0 | 22 | 7 | 29 | | |
| | Total | 169 | 57 | 13 | 239 | 101.556 | 0.000 |
| No | Total | 45 | 0 | 0 | 45 | | |

Fisher's exact test: $\chi^2$ = 23.575, P = 0.000<0.01.

Kruskal-Wallis test: $\chi^2$ = 12.971, P = 0.002<0.01

a: the risk value is 28.27

b: the risk value is 39.38

c: the risk value is 48.50.

## The AI inflammation score

The AI inflammation score was determined based on the quantitative evaluation of lung inflammation by AI-assisted chest HRCT and was further used to analyze its relationship with the degree of pulmonary fibrosis. The analysis revealed significant differences in the degree of pulmonary inflammation (PIV, PIV/WLV) and the extent of the affected area (the affected lung segments and lobes) among the three groups (P<0.05 or 0.01, Fig 1A). These results confirmed that there were significant differences in the extent and degree of lung inflammation among patients with mild, moderate and severe pulmonary fibrosis; that is, patients with severe pulmonary fibrosis had the most severe and extensive lung inflammation, followed by patients with moderate pulmonary fibrosis, while patients with mild pulmonary fibrosis had the least severe and extensive lung inflammation.

## Risk factors for pulmonary fibrosis

After stratifying the 239 patients with pulmonary fibrosis during hospitalization according to their degree of pulmonary fibrosis at discharge, their clinical characteristics were analyzed. Patients with or without pulmonary fibrosis had statistically significant differences in age, IL-6 levels, lymphocyte %, aspartate transaminase (AST), albumin, CRP/albumin ratio, platelet/

**Table 2. The type of lesions and the affected lung segments in 239 COVID-19 patients with pulmonary fibrosis at discharge.**

| Affected lung segments | Mild[a] (n = 114) | Moderate[b] (n-96) | Severe[c] (n = 29) | Z value, P value | | |
|---|---|---|---|---|---|---|
| | | | | a vs. b | a vs. c | b vs. c |
| **Ground-glass opacity [n (%)]** | 92(80.70%) | 72(75.00%) | 28(96.55%) | | | |
| Affected lung segments | 4.80±2.80 | 6.29±3.64 | 9.50±3.97 | -2.681, 0.007 | -5.572, 0.000 | -3.685, 0.000 |
| **Linear opacities [n (%)]** | 94(82.46%) | 91(94.79%) | 29(100.00%) | | | |
| Affected lung segments | 3.33±1.89 | 4.75±2.48 | 6.00±2.94 | -4.072, 0.000 | -4.804, 0.000 | -2.206, 0.027 |
| **Interlobular septal thickening [n (%)]** | 10(8.77%) | 72(75.00%) | 29(100.00%) | | | |
| Affected lung segments | 2.70±1.64 | 3.88±1.58 | 6.03±3.36 | -2.571, 0.10 | -3.399, 0.001 | -2.968, 0.003 |
| **Reticulation [n (%)]** | 4(3.51%) | 39(40.63%) | 28(96.55%) | | | |
| Affected lung segments | 1.25±0.50 | 3.64±2.28 | 5.39±2.42 | -2.707, 0.007 | -3.1296, 0.001 | -3.190, 0.001 |
| **Honeycombing or bronchiectasis [n (%)]** | 2(1.75%) | 55(57.29%) | 29(100.00%) | | | |
| Affected lung segments | 1.00 | 2.82±1.56 | 4.41±2.73 | — | -2.657, 0.008 | — |
| **Pulmonary fibrosis score** | 5.53±2.21 | 14.81±2.49 | 24.17±1.54 | -12.513, 0.000 | -8.362, 0.000 | -8.176, 0.000 |

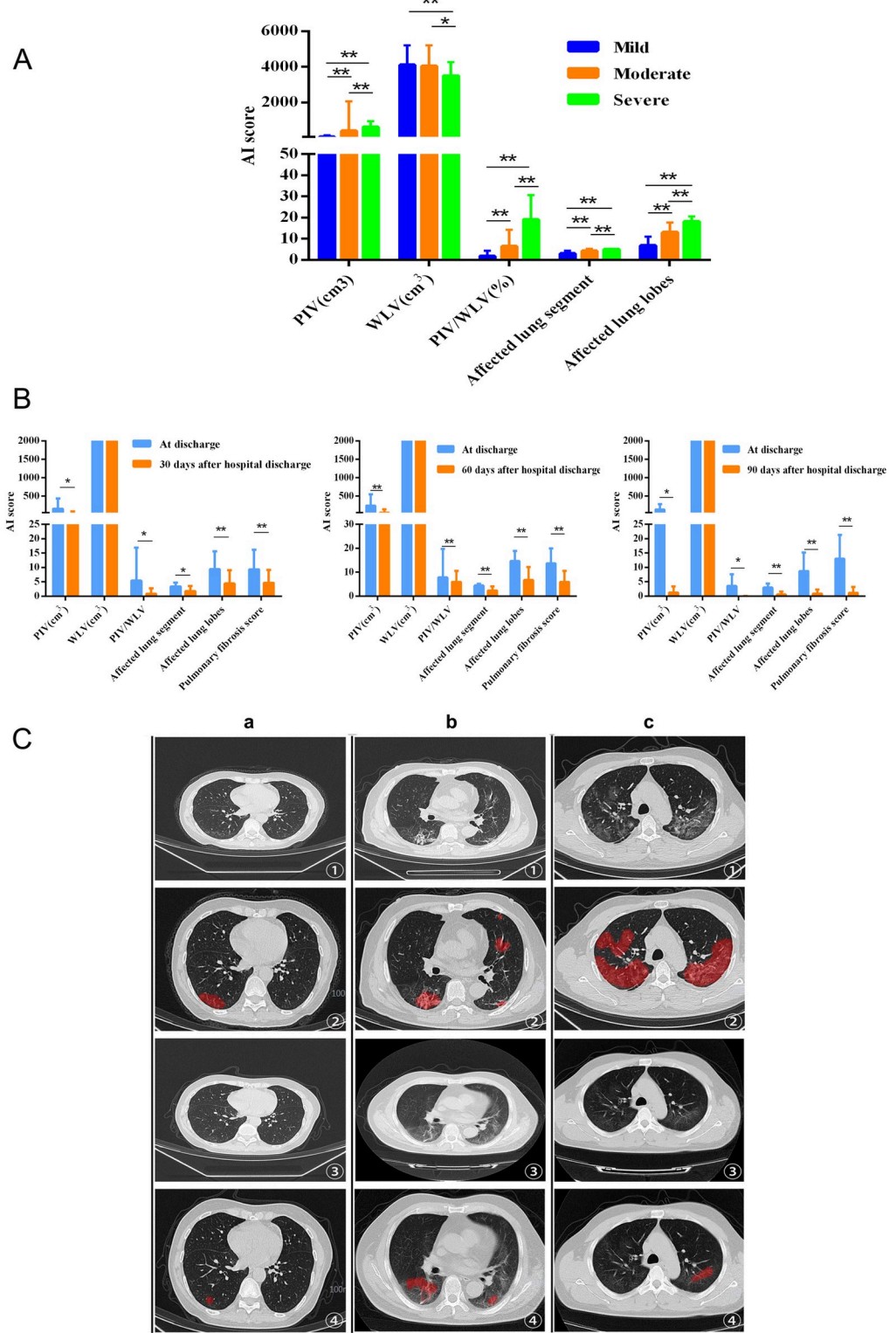

**Fig 1.** A. The AI inflammation scores of 239 COVID-19 patients with pulmonary fibrosis at discharge. *, P<0.05; **, P<0.01; B. Comparison of the AI inflammation score or pulmonary fibrosis score between patients at discharge and 30 days (n = 34 patients), 60 days (n = 11 patients) and 90 days (n = 7 patients) after discharge. C. Comparison of chest HRCT results between patients at discharge and 30 days after hospital discharge. a, b and c represent three separate patients: a represents a patient with mild pulmonary fibrosis, b represents a patient with moderate pulmonary fibrosis,

and c represents a patient with severe pulmonary fibrosis. ① represents the chest HRCT at discharge; ② represents the extent of lesions marked by AI at discharge (red); ③ represents the chest HRCT 30 days after hospital discharge; ④ represents the extent of lesions marked by AI 30 days after hospital discharge (red).

lymphocyte ratio and some other indexes (Table 3), suggesting that these abnormal clinical indicators may be related to the pulmonary fibrosis.

## Analysis of risk factors in patients with pulmonary fibrosis

Multivariate binary logistic regression analysis was used to evaluate the relationship between the quantitative pulmonary fibrosis score and the related risk factors in 248 COVID-19 patients (Table 4). Our results showed that there were significant relationships between pulmonary fibrosis and the levels of albumin and IL-6, which suggests that IL-6 and albumin are independent risk factors affecting pulmonary fibrosis. The regression equation is $logit(P) = \beta_0 + \beta_1{}^*X_1 + \beta_2{}^*X_2$ (where $\beta_0 = 9.964$, $\beta_1 = 0.078$, $\beta_2 = -0.197$, $X_1$ is the IL-6 level in the acute stage, and $X_2$ is the albumin level).

## Prognosis of patients with pulmonary fibrosis

Thirty-four of the 239 COVID-19 patients underwent chest HRCT 30 days after hospital discharge (23 moderate, 7 severe, 4 critical), 11 patients underwent chest HRCT 60 days after discharge (3 moderate, 6 severe, 2 critical), and 7 patients underwent chest HRCT 90 days after discharge (4 moderate, 1 severe, 2 critical). All of these patients showed significant improvements in ground-glass opacities, linear opacities, interlobular septal thickening, reticulation, honeycombing and bronchiectasis (Table 5, P<0.01). The linear opacities, interlobular septal thickening, reticulation, honeycombing and bronchiectasis were completely resolved 90 days after discharge.

## AI inflammation score as an evaluation of the evolution of pulmonary fibrosis

The AI inflammation score and quantitative pulmonary fibrosis score in the patients included in this study were analyzed 30 days, 60 days and 90 days after hospital discharge. The results showed that the degree of pulmonary inflammation (PIV, PIV/WLV), the extent of the affected area (the affected lung segment and lobes), and the quantitative pulmonary fibrosis score were significantly improved 30, 60 and 90 days after discharge (Fig 1B and 1C, P<0.01).

## Discussion

Quantitative CT scores have been used to assess the severity of pulmonary fibrosis in patients with idiopathic pulmonary fibrosis [16–19]. In this study, we found that approximately 80% of the 284 COVID-19 patients had pulmonary fibrosis at discharge. The incidence of pulmonary fibrosis in patients with moderate COVID-19 was relatively low (incidence rate 73.8%), but it occurred in all patients with severe or critical COVID-19. In particular, patients with moderate or severe COVID-19 usually developed mild-to-moderate pulmonary fibrosis, while patients with critical COVID-19 generally developed severe pulmonary fibrosis. These results confirmed that the degree of fibrosis was more severe in patients with critical COVID-19 than in patients with moderate or severe COVID-19, which may be related to the fact that acute-stage lung inflammation was more extensive and severe in patients with severe and critical cases of COVID-19.

**Table 3. The clinical characteristics of COVID-19 patients with pulmonary fibrosis at discharge.**

| | Patients with pulmonary fibrosis | Patients without pulmonary fibrosis | $\chi^2$/T value | P value |
|---|---|---|---|---|
| **Number** | 239 | 45 | — | — |
| **Age, years, mean (SD)** | 55.87±1.03 | 47.29±2.85 | 3.216 | 0.002※ |
| **Sex** | | | | |
| Female | 136 | 26 | 0.012 | 0.914 |
| Male | 103 | 19 | | |
| **Symptom** | | | | |
| High fever (≥38.5°C) | 99 | 7 | 10.83 | 0.001※ |
| Cough | 164 | 24 | 3.954 | 0.047# |
| Wheezing | 69 | 8 | 2.358 | 0.125 |
| Chest tightness | 66 | 6 | 4.082 | 0.043# |
| **IL-6** | | | | |
| Acute stage | 30.86±2.58 | 5.99±1.00 | 4.163 | 0.000※ |
| Hospital discharge | 5.89±0.53 | 2.59±0.53 | 3.577 | 0.000※ |
| **Laboratory findings** | | | | |
| WBC, ×10⁹ per L | 5.56±0.18 | 5.51±0.22 | 0.102 | 0.919 |
| Lymphocyte ×10⁹ per L | 1.24±0.07 | 1.70±0.11 | 2.583 | 0.010※ |
| Lymphocyte % | 24.58±0.81 | 31.43±1.76 | 3.381 | 0.001※ |
| Platelet, ×10⁹ per L | 200.10±4.90 | 212.80±9.60 | 1.054 | 0.293 |
| HB (g/L) | 127.10±1.16 | 132.20±2.65 | 1.767 | 0.078 |
| CRP (mg/L) | 28.00±2.52 | 7.38±1.36 | 3.534 | 0.001※ |
| PCT (ng/mL) | 0.31±0.14 | 0.04±0.00 | 0.808 | 0.420 |
| ALT, U/L | 33.42±2.63 | 25.58±4.28 | 1.236 | 0.217 |
| AST, U/L | 35.52±1.44 | 26.78±2.18 | 2.531 | 0.011※ |
| Albumin (g/L) | 38.34±0.31 | 42.28±0.51 | 5.270 | 0.000※ |
| Creatinine (μmol/L) | 67.33±1.22 | 67.82±2.23 | 0.166 | 0.868 |
| Glucose (mmol/L) | 6.74±0.17 | 6.13±0.34 | 1.421 | 0.156 |
| Potassium (mmol/L) | 3.86±0.03 | 3.91±0.07 | 0.620 | 0.536 |
| CK (U/L) | 68.11±10.91 | 54.09±9.05 | 0.547 | 0.585 |
| Myoglobin (μg/L) | 54.25±4.93 | 42.21±7.71 | 1.016 | 0.311 |
| hs-cTnT, pg/mL | 0.07±0.04 | 0.01±0.00 | 0.640 | 0.522 |
| Prothrombin time, s | 12.08±0.08 | 15.38±2.50 | 3.031 | 0.003※ |
| APTT | 32.42±0.25 | 33.56±0.53 | 1.822 | 0.070 |
| D-dimer (mg/L) | 387.80±69.36 | 164.70±40.40 | 1.386 | 0.167 |
| AST/ALT | 1.38±0.64 | 1.33±0.51 | -0.067 | 0.946 |
| CRP/albumin ratio | 0.80±1.19 | 0.18±0.22 | -4.946 | 0.000※ |
| Platelet/lymphocyte ratio | 209.98±148.92 | 146.72±70.49 | -3.151 | 0.002# |
| **Cellular immunity-related indexes** | | | | |
| CD3 count (N = 174+37) | 790.90±33.23 | 1262.00±89.02 | 5.687 | 0.000※ |
| CD4 count (N = 174+37) | 451.10±21.24 | 735.70±57.62 | 5.355 | 0.000※ |
| CD8 count (N = 174+37) | 282.00±15.01 | 418.40±36.19 | 3.728 | 0.000※ |
| CD19 count (N = 174+37) | 160.60±7.57 | 277.20±31.01 | 5.371 | 0.000※ |
| CD16+56 (N = 174+37) | 216.60±16.37 | 246.40±35.07 | 0.765 | 0.445 |
| **Humoral immunity-related indexes** | | | | |
| IgG (g/L) (N = 112+17) | 9.79±0.57 | 9.54±1.84 | 0.153 | 0.879 |
| IgM (g/L) (N = 112+17) | 4.81±0.59 | 6.14±1.67 | 0.815 | 0.417 |
| IgA (g/L) (N = 112+17) | 2.64±0.90 | 1.62±0.20 | 0.438 | 0.662 |

#, P<0.05

※, P<0.01

**Table 4. Analysis of risk factors in patients with pulmonary fibrosis.**

| Index | HR | 95%CI | P value |
|---|---|---|---|
| Age(years) | 1.001 | 0.975–1.027 | 0.941 |
| High fever (≥38.5˚C) | 0.502 | 0.174–1.450 | 0.203 |
| Cough | 0.708 | 0.309–1.621 | 0.708 |
| Chest tightness | 0.772 | 0.263–2.270 | 0.757 |
| IL-6(acute stage)* | 1.081 | 1.021–1.144 | 0.007 |
| IL-6 (hospital discharge) | 1.119 | 0.969–1.292 | 0.125 |
| Lymphocyte ×$10^9$ per L | 0.921 | 0.711–1.194 | 0.536 |
| Lymphocyte % | 0.988 | 0.955–1.022 | 0.479 |
| AST (U/L) | 1.02 | 0.990–1.051 | 0.192 |
| Albumin (g/L) * | 0.821 | 0.734–0.918 | 0.001 |
| PT (s) | 0.93 | 0.822–1.052 | 0.250 |

AI-assisted chest HRCT technology was used to quantitatively evaluate the extent and degree of lung inflammation, and it provided a relatively accurate evaluation of the inflammatory status of the lungs. We observed that patients with severe pulmonary fibrosis had more extensive and severe lung inflammation than those with mild or moderate fibrosis. The AI inflammation score demonstrated good consistency with the quantitative pulmonary fibrosis score, which showed that AI-assisted chest HRCT technology could be used to not only quantify the degree of lung inflammation but also indirectly reflect the degree of pulmonary fibrosis, providing qualitative and quantitative data that could be used to analyze the long-term evolution of pulmonary fibrosis.

Previous studies reported that IL-6 could serve as an indicator of the progression of COVID-19 [20]. IL-6 levels can reflect the severity of the inflammatory response. Our results showed that IL-6 is an independent risk factor for pulmonary fibrosis. Its underlying mechanisms may be as follows: IL-6 activates neutrophils and promotes their accumulation at the

**Table 5. The prognosis of patients with pulmonary fibrosis.**

| The affected lung segment | Follow-up after 30 days (n = 34) | | | Follow-up after 60 days (n = 11) | | | Follow-up after 90 days (n = 7) | | |
|---|---|---|---|---|---|---|---|---|---|
| | At discharge | After 30 days | t/Z, P | At discharge | After 60 days | t/Z, P | At discharge | After 90 days | t/Z, P |
| **Ground-glass opacity [n (%)]** | 29(85.29%) | 18(52.94%) | | 10(90.91%) | 4(36.36%) | | 5(71.43%) | 1(14.29%) | |
| Affected lung segments | 5.74±4.31 | 3.26±3.73 | -2.427, 0.015 | 8.27±4.45 | 3.36±3.36 | -2.494, 0.13 | 4.29±5.06 | 0.43±1.13 | -1.986, 0.47 |
| **Linear opacities [n (%)]** | 27(79.41%) | 20(58.82%) | | 11 (100.00%) | 8(72.73%) | | 7(100.00%) | 2(28.57%) | |
| Affected lung segments | 3.18±2.37 | 2.06±2.51 | -2.175, 0.030 | 5.00±2.35 | 2.45±2.88 | -2.124, 0.034 | 3.71±1.80 | 0.71±1.25 | -2.688, 0.07 |
| **Interlobular septal thickening [n (%)]** | 10(29.41%) | 4(11.76%) | | 4(36.36%) | 2(18.18%) | | 3(42.86%) | 0(0.00%) | |
| Affected lung segments | 1.15±2.16 | 0.32±0.94 | -1.8678,0.062 | 1.45±2.34 | 0.64±1.57 | -0.963,0.336 | 1.29±1.89 | 0.00 | — |
| **Reticulation [n (%)]** | 8(23.53%) | 1(2.94%) | | 4(36.36%) | 1(9.09%) | | 5(71.43%) | 0(0.00%) | |
| Affected lung segments | 0.82±1.68 | 0.09±0.51 | -2.499,0.012 | 1.18±2.32 | 0.27±0.90 | -1.122,0.262 | 1.29±1.89 | 0.00 | — |
| **Honeycombing or bronchiectasis [n (%)]** | 8(23.53%) | 2(5.88%) | | 4(36.36%) | 1(9.09%) | | 2(28.57%) | 0(0.00%) | |
| Affected lung segments | 0.71±1.75 | 1.65±4.48 | -4.253,0.000 | 1.55±1.44 | 0.09±0.30 | -2.749, 0.006 | 1.71±3.40 | 0.00 | — |

site of injury, induces the release of protease and oxygen free radicals, and finally leads to pulmonary interstitial edema and a severe inflammatory response [21]. Moreover, we also identified albumin as an independent risk factor for pulmonary fibrosis, which emphasized the importance of correcting albumin abnormalities in patients with COVID-19.

It was reported that the AST/ALT ratio on admission was significantly associated with in-hospital mortality in COVID-19 patients [22]. In this work, we found that there were significant relationships between the levels of AST and pulmonary fibrosis. AST displays the highest activity in the liver and skeletal muscle but also occurs in several tissues, including lungs, heart muscle, kidneys, pancreas, brain, leucocyte and erythrocytes. AST is less specific for liver damage compared to ALT [22]. Therefore, COVID-19 patients with a significant increase in AST may also have large damage in other tissues, including the lungs, which may affect pulmonary fibrosis at discharge. Larger studies are required to confirm the capacity of this parameter to independently predict pulmonary damage and fibrosis in this group. We have revised it and added the article as a reference in the revised version.

In addition, we found that the degree of pulmonary inflammation (PIV, PIV/WLV), the extent of the affected area (the affected lung segments and lobes), and the pulmonary fibrosis scores of COVID-19 patients were significantly improved 30 days, 60 days and 90 days after discharge, which confirmed that pulmonary fibrosis was likely to resolve after discharge. Meanwhile, we also noted that the pulmonary fibrosis in some patients did not completely resolve within 90 days, and whether additional antifibrosis treatment could accelerate the process is worthy of further investigation. A proportion of SARS patients still had obvious additional improvements in interstitial injuries and lung functional recovery 2 years after hospital discharge [23]. Therefore, it is theoretically possible for patients with incompletely resolved pulmonary fibrosis within 90 days to continue to recover.

As it was retrospective, this study has some limitations. First, many of the patients were unwilling to undergo periodic HRCT reexaminations after discharge, and the relatively few patients with follow-up data limited our analysis and led to biased results. Second, pulmonary function data were unavailable in this study, and the extent to which pulmonary fibrosis influences pulmonary function remains unclear; however, our study explored the dynamic evolution of the structural abnormalities in the lungs using HRCT, and lung function should theoretically be restored if the lung structure returns to normal. Third, most of the patients admitted to our hospital were had moderate, severe or critical COVID-19, so it is still unknown whether pulmonary fibrosis occurs in patients with mild COVID-19 and, if so, what the clinical characteristics are; however, the study site was a designated hospital that treated a large number of COVID-19 patients in Wuhan, and our study included all patients who achieved a clinical cure in our hospital within 2 months. Therefore, this study is still representative of pulmonary fibrosis in COVID-19 patients.

In conclusion, AI-assisted chest HRCT technology was used in this study, and we found that the more severe the clinical classification of COVID-19 was, the more extensive and severe the lung inflammation and the more severe the residual pulmonary fibrosis. In most of these patients, pulmonary fibrosis improved or even resolved within 90 days after discharge. Furthermore, the levels of IL-6 and albumin are independent risk factors for pulmonary fibrosis and should be regarded as important therapeutic targets for the treatment of COVID-19 patients with pulmonary fibrosis.

## Supporting information

**S1 File.**
(XLSX)

**S2 File.**
(XLSX)

**S3 File.**
(XLSX)

**S4 File.**
(XLSX)

## Author Contributions

**Conceptualization:** Shi-Ming Chen.

**Data curation:** Jia-Ni Zou, Liu Sun, Bin-Ru Wang, You Zou.

**Formal analysis:** Shan Xu, Yong-Jun Ding.

**Investigation:** Li-Jun Shen, Wen-Cai Huang.

**Methodology:** Xiao-Jing Jiang.

**Writing – original draft:** Bin-Ru Wang, You Zou.

**Writing – review & editing:** Shi-Ming Chen.

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
