## [Decision Letter · Decision Letter 0]

9 Dec 2020

PONE-D-20-31950

The characteristics and evolution of pulmonary fibrosis in COVID-19 patients as assessed by AI-assisted chest HRCT

PLOS ONE

Dear Dr. Chen,

Thank you for submitting your manuscript to PLOS ONE. After careful consideration, we feel that it has merit but does not fully meet PLOS ONE’s publication criteria as it currently stands. Therefore, we invite you to submit a revised version of the manuscript that addresses the points raised during the review process.

We look forward to receiving your revised manuscript.

Kind regards,

Giordano Madeddu

Academic Editor

PLOS ONE

Journal Requirements:

2. In the ethics statement in the manuscript and in the online submission form, please provide additional information about the patient records/samples used in your retrospective study, including: a) whether all data were fully anonymized before you accessed them; b) the date range (month and year) during which patients' medical records/samples were accessed.

"This study was supported by grant 81770981 and 82002863 from the

National Natural Science Foundation of China and grant [2018]116 from Wuhan

Municipality Young and Middle-aged Medical Talent Cultivation Program."

"Unfunded studies"

Reviewers' comments:

Reviewer's Responses to Questions

**Comments to the Author**

1. Is the manuscript technically sound, and do the data support the conclusions?

Reviewer #1: Yes

Reviewer #2: Yes

2. Has the statistical analysis been performed appropriately and rigorously? 

Reviewer #1: Yes

Reviewer #2: Yes

3. Have the authors made all data underlying the findings in their manuscript fully available?

Reviewer #1: Yes

Reviewer #2: Yes

4. Is the manuscript presented in an intelligible fashion and written in standard English?

Reviewer #1: No

Reviewer #2: Yes

5. Review Comments to the Author

Reviewer #1: Dear Authors,

First, I would like to thank you for the paper you submitted. After the great interest grown in the scientific community in the first months of the year toward the acute phase of COVID-19 and its possible treatments, now it is the time to ask us what will happen after the infection, and if these modifications will affect survivors for the rest of their lives. In the wake of the outcomes of SARS-CoV and MERS-CoV pneumonia, many papers investigating on the possible long-time effects of COVID-19 have been published in the past few months, but, by now, the follow-up timing is too short to establish a permanent effect. During 2020 numerous studies investigated toward the role of AI in the diagnosis of COVID-19 pneumonia, but this is possibly the first paper published on the use of AI in the evolution of pneumonia after the clinical healing and its role in predicting fibrosis, even if there are many limitations, that you correctly identified.

However, I have a few concerns I will enlist you as follow.

Minor concerns:

- In the abstract, you repeated the phrase “IL-6 and albumin are independent risk factors for pulmonary fibrosis” twice.

- Page 4, line 66: there is a typo, for it is written “SARA-CoA-2” instead of “SARS-CoV-2”.

- Page 5, line 88: milliampere-second is written in one word.

- Page 6, line 99: “anterior endobasal segment” is not a term commonly used in the anatomy of lung segments. Did you mean medial segment of the lower lobe? Please, correct.

- Page 8, line 149: you say that 248 patients were enrolled, clearly a refuse.

Major concerns:

- Page 3, line 33-34: there is an imprecision that I would suggest you to correct. In fact, you correctly reported that SARS-CoV-2 is similar for the 85% of genome to a bat coronavirus, but after that you included on parenthesis SARS-CoV: SARS-CoV-2, instead, share 85% of its genome with bat coronavirus bat-SL-CoVZC45 (Xu J, Zhao S, Teng T, Abdalla AE, Zhu W, Xie L, Wang Y, Guo X. Systematic Comparison of Two Animal-to-Human Transmitted Human Coronaviruses: SARS-CoV-2 and SARS-CoV. Viruses. 2020 Feb 22;12(2):244. doi: 10.3390/v12020244. PMID: 32098422; PMCID: PMC7077191).

- Page 6, line 95-96: you used the term “shadow” to describe many CT findings associated with COVID-19 pneumonia. This term it is not used by other radiologists in an international setting: I suggest you to change it with the common used “linear opacities” and honeycombing. I did not understand the term “mesh shadows”: could you explain it better?

- In page 6, line 104, you cited a score for lung fibrosis proposed by Camiciottoli, followed by an apex “1”. You didn’t cite the relative paper.

- In page 8, line 150 you say that patients with pulmonary fibrosis were 239, while in the rest of the text you say 237. I suggest you to correct the refuse.

- Page 9, paragraph “The relationship between the degree of pulmonary fibrosis and the clinical classification”: I suggest you to write again the initial part of the paragraph where you enlist the amount of patients with fibrosis for each group, because as it is written by now it is confusing and it is necessary to look at table 1 to understand what you meant.

- In the study, you found a statistical association with the elevation of AST and severity of fibrosis. A recently published study showed a peculiar association between AST/ALT ratio, known also as de Ritis ratio, and hospital mortality in COVID-19 patients, tied also to a larger pneumonia (Zinellu A, Arru F, De Vito A, Sassu A, Valdes G, Scano V, Zinellu E, Perra R, Madeddu G, Carru C, Pirina P, Mangoni AA, Babudieri S, Fois AG. The De Ritis ratio as prognostic biomarker of in-hospital mortality in COVID-19 patients. Eur J Clin Invest. 2020 Oct 11:e13427. doi: 10.1111/eci.13427. Epub ahead of print. PMID: 33043447; PMCID: PMC7646002). I suggest you to read this work and evaluate if de Ritis ratio was also associated with more severe pulmonary fibrosis. Moreover, another study found an association with others laboratory findings and combined indexes, and I suggest you to have a look for a comparison: Paliogiannis P, Zinellu A, Scano V, Mulas G, De Riu G, Pascale RM, Arru LB, Carru C, Pirina P, Mangoni AA, Fois AG. Laboratory test alterations in patients with COVID-19 and non COVID-19 interstitial pneumonia: a preliminary report. J Infect Dev Ctries. 2020 Jul 31;14(7):685-690. doi: 10.3855/jidc.12879. PMID: 32794454.

- It is a great pity that after all the good analysis you conducted you didn’t specify the category of patients who underwent CT at 30, 60 and 90 days from discharge: it would be very interesting to know if they were all severe cases or belonged to different groups. I suggest you to specify it. I would also like to know why only few patients had a new CT scan: were they symptomatic? Did symptoms mismatch CT findings? Why you suggest the use pf antibiotics in patients showing residual interstitial abnormalities after 90 days?

The abstract is concise and it is representative of the full article, and discussion and conclusion are coherent with the study and data shown.

The paper is generally well-written, but there are many phrase constructions atypical for English grammar which probably reflects a literal translation from your native language. I suggest an overall revision of the language, preferably from a mother tongue one.

Reviewer #2: General comments

The paper entitled “The characteristics and evolution of pulmonary fibrosis in COVID-19 patients as assessed by AI-assisted chest HRCT” by Chen and Co-wokers is aimed at evaluating a possible role of artificial intelligence applied to chest CT in order to provide an important basis for the clinical diagnosis, treatment and prognosis of COVID-19 pulmonary fibrosis. The aim is clear and the topic is interesting and innovative. The paper is well written. Some shortcomings need to be addressed; in particular, the retrospective nature of this study and the relative high number of missed CT could affect the inference of the results.

Detailed comments

Introduction:

- “Pulmonary fibrosis is a serious complication of viral pneumonia, which often leads to dyspnea and impaired lung function”: I suggest “pulmonary fibrosis can occur as a serious …”

- Line 43: I suggest: “Chest x-rays and high-resolution-CT (citing Severity of lung involvement on chest X-rays in SARS-coronavirus-2 infected patients as a possible tool to predict clinical progression: an observational retrospective analysis of the relationship between radiological, clinical, and laboratory data. Baratella E, Crivelli P, Marrocchio C, Bozzato AM, Vito A, Madeddu G, Saderi L, Confalonieri M, Tenaglia L, Cova MA. J Bras Pneumol. 2020 Sep 21;46(5):e20200226. doi: 10.36416/1806-3756/e20200226. eCollection 2020).

- Methods:

- Chest CT protocol: Authors could better explain the mean radiation dose exposure during CT.

- Chest CT image analysis: Authors could better clarify the years of experience on chest CT images.

- “… striated shadow or mesh shadow…” Authors could explain the mean of these words.

6. PLOS authors have the option to publish the peer review history of their article (what does this mean?). If published, this will include your full peer review and any attached files.

Reviewer #1: No

Reviewer #2: No

---

## [Author Response · Author response to Decision Letter 0]

11 Jan 2021

Dec 22, 2020

Dear Editors and Reviewers:

Thank you for your letter and for the reviewers’ comments concerning our manuscript entitled “The characteristics and evolution of pulmonary fibrosis in COVID-19 patients as assessed by AI-assisted chest HRCT”. (ID: PONE-D-20-31950).

These comments are very constructive and helpful for revising our paper. We have discussed the comments carefully and have made revisions which we hope will be met with approval. They are all marked in blue in the revised version. The main answers to the reviewer’s comments are as follow: 

1.Please ensure that your manuscript meets PLOS ONE's style requirements, including those for file naming.

Answer: Thank you for this comment. We have revised the article format according to PLOS ONE's style requirements.

2.In the ethics statement in the manuscript and in the online submission form, please provide additional information about the patient records/samples used in your retrospective study, including: a) whether all data were fully anonymized before you accessed them; b) the date range (month and year) during which patients' medical records/samples were accessed. 

Answer: Thank you for this comment. We make sure that all data were anonymous. The patients' medical records were accessed from February 1 to June 8, 2020. We have added these to the revised version and marked in blue in page 5, line 77-78.

3.Thank you for stating the following in the Funding Section of your manuscript:

"This study was supported by grant 81770981 and 82002863 from the

National Natural Science Foundation of China and grant [2018]116 from Wuhan

Municipality Young and Middle-aged Medical Talent Cultivation Program."

"Unfunded studies"

Answer: Thank you for this comment. We have removed the fund information in the manuscript and added it in the cover letter. However, we can not added it in the online submission form. We would be very grateful if you could change the online submission form on our behalf.

4.PLOS requires an ORCID iD for the corresponding author in Editorial Manager on papers submitted after December 6th, 2016. Please ensure that you have an ORCID iD and that it is validated in Editorial Manager. To do this, go to ‘Update my Information’ (in the upper left-hand corner of the main menu), and click on the Fetch/Validate link next to the ORCID field. This will take you to the ORCID site and allow you to create a new iD or authenticate a pre-existing iD in Editorial Manager. Please see the following video for instructions on linking an ORCID iD to your Editorial Manager account: https://www.youtube.com/watch?v=_xcclfuvtxQ

Answer: Thank you for this comment. We have already applied ORCID iD (0000-0002-1216-004X).

5.Your ethics statement should only appear in the Methods section of your manuscript. If your ethics statement is written in any section besides the Methods, please delete it from any other section.

Answer: Thank you for this comment. We've removed ethics statement in any other section.

Reviewer 1:

Specific comments:

1. In the abstract, you repeated the phrase “IL-6 and albumin are independent risk factors for pulmonary fibrosis” twice.

Answer: Thank you for this comment. We have revised it and marked in blue in page 2, line 12. 

2. there is a typo, for it is written “SARA-CoA-2” instead of “SARS-CoV-2”.

Answer: Thank you for this comment. We have revised it and marked in blue in page 5, line 69. 

3. milliampere-second is written in one word.

Answer: Thank you for this comment. We have revised it and marked in blue in page 6, line 89-90. 

4.“anterior endobasal segment” is not a term commonly used in the anatomy of lung segments. Did you mean medial segment of the lower lobe? Please, correct.

Answer: Thank you for this comment. “anterior endobasal segment” mean“anterior and basal segmental”. We have revised it and marked in blue in page 6, line 97-99. 

5.you say that 248 patients were enrolled, clearly a refuse.

Answer: Thank you for this comment. We have revised it and marked in blue in page 9, line 157. 

6. there is an imprecision that I would suggest you to correct. In fact, you correctly reported that SARS-CoV-2 is similar for the 85% of genome to a bat coronavirus, but after that you included on parenthesis SARS-CoV: SARS-CoV-2, instead, share 85% of its genome with bat coronavirus bat-SL-CoVZC45 (Xu J, Zhao S, Teng T, Abdalla AE, Zhu W, Xie L, Wang Y, Guo X. Systematic Comparison of Two Animal-to-Human Transmitted Human Coronaviruses: SARS-CoV-2 and SARS-CoV. Viruses. 2020 Feb 22;12(2):244. doi: 10.3390/v12020244. PMID: 32098422; PMCID: PMC7077191).

Answer: Thank you for this comment. We have revised SARS-CoV-2 share 85% of its genome with bat coronavirus bat-SL-CoVZC45 in the revised version and marked in blue in page 3, line 34-35. 

7. you used the term “shadow” to describe many CT findings associated with COVID-19 pneumonia. This term it is not used by other radiologists in an international setting: I suggest you to change it with the common used “linear opacities” and honeycombing. I did not understand the term “mesh shadows”: could you explain it better?

Answer: Thank you for this comment. “shadow”should be “linear opacities”. “mesh shadows”should be “reticulation”. We have revised it and marked in blue in the Manuscript and Revised Manuscript with Track Changes. 

8. you cited a score for lung fibrosis proposed by Camiciottoli, followed by an apex “1”. You didn’t cite the relative paper.

Answer: Thank you for this comment. We have added references in the Manuscript and Revised Manuscript with Track Changes. (Camiciottoli G, Orlandi I, Bartolucci M, Meoni E, Nacci F, Diciotti S, et al Lung CT densitometry in systemic sclerosis: correlation with lung function, exercise testing, and quality of life. Chest. 2007;131(3):672-681. doi.10.1378/chest.06-1401. )

9. you say that patients with pulmonary fibrosis were 239, while in the rest of the text you say 237. I suggest you to correct the refuse.

Answer: Thank you for this comment. The patients with pulmonary fibrosis were 239. We have corrected it in the rest of the text and marked in blue in page 9, line 173.

10. paragraph “The relationship between the degree of pulmonary fibrosis and the clinical classification”: I suggest you to write again the initial part of the paragraph where you enlist the amount of patients with fibrosis for each group, because as it is written by now it is confusing and it is necessary to look at table 1 to understand what you meant.

Answer: Thank you for this comment. We have write again the paragraph and marked in blue in page 9-10, line 173-182.

11.In the study, you found a statistical association with the elevation of AST and severity of fibrosis. A recently published study showed a peculiar association between AST/ALT ratio, known also as de Ritis ratio, and hospital mortality in COVID-19 patients, tied also to a larger pneumonia (Zinellu A, Arru F, De Vito A, Sassu A, Valdes G, Scano V, Zinellu E, Perra R, Madeddu G, Carru C, Pirina P, Mangoni AA, Babudieri S, Fois AG. The De Ritis ratio as prognostic biomarker of in-hospital mortality in COVID-19 patients. Eur J Clin Invest. 2020 Oct 11:e13427. doi: 10.1111/eci.13427. Epub ahead of print. PMID: 33043447; PMCID: PMC7646002). I suggest you to read this work and evaluate if de Ritis ratio was also associated with more severe pulmonary fibrosis. Moreover, another study found an association with others laboratory findings and combined indexes, and I suggest you to have a look for a comparison: Paliogiannis P, Zinellu A, Scano V, Mulas G, De Riu G, Pascale RM, Arru LB, Carru C, Pirina P, Mangoni AA, Fois AG. Laboratory test alterations in patients with COVID-19 and non COVID-19 interstitial pneumonia: a preliminary report. J Infect Dev Ctries. 2020 Jul 31;14(7):685-690. doi: 10.3855/jidc.12879. PMID: 32794454.

Answer: Thank you for this comment. It is a very good suggestion.We have found that CRP/albumin ratio, platelet/ lymphocyte ratio was associated with pulmonary fibrosis. We have included these results in the revised version and marked in blue in page 13. 

12.It is a great pity that after all the good analysis you conducted you didn’t specify the category of patients who underwent CT at 30, 60 and 90 days from discharge: it would be very interesting to know if they were all severe cases or belonged to different groups. I suggest you to specify it. I would also like to know why only few patients had a new CT scan: were they symptomatic? Did symptoms mismatch CT findings? Why you suggest the use pf antibiotics in patients showing residual interstitial abnormalities after 90 days?

The abstract is concise and it is representative of the full article, and discussion and conclusion are coherent with the study and data shown. The paper is generally well-written, but there are many phrase constructions atypical for English grammar which probably reflects a literal translation from your native language. I suggest an overall revision of the language, preferably from a mother tongue one.

Answer: Thank you for these comment. 

They belonged to different groups.We have revised it and marked in blue in page 14 line 238-240

All of the patients are required to undergo a chest CT scan after discharge in accordance with the guidelines, but many of them worry about radiation in chest CT. So only few patients had a new CT scan. And their symptoms were matched with CT findings.

As for the usage of antibiotics, we have made a mistake. It should be anti-fibrosis treatment. We have revised it and marked in blue in page 17, line 286. 

As for the English grammar, we also know this is a very important issue. The revised manuscript has been proofread by an English-speaking professional with science background at Springer Nature Author Services. The certificate is uploaded as an attachment.

Reviewer 2:

1.“Pulmonary fibrosis is a serious complication of viral pneumonia, which often leads to dyspnea and impaired lung function”: I suggest “pulmonary fibrosis can occur as a serious …”

Answer: Thank you for this comment.We have revised it and marked in blue in page 3, line 24 .

2. I suggest: “Chest x-rays and high-resolution-CT (citing Severity of lung involvement on chest X-rays in SARS-coronavirus-2 infected patients as a possible tool to predict clinical progression: an observational retrospective analysis of the relationship between radiological, clinical, and laboratory data. Baratella E, Crivelli P, Marrocchio C, Bozzato AM, Vito A, Madeddu G, Saderi L, Confalonieri M, Tenaglia L, Cova MA. J Bras Pneumol. 2020 Sep 21;46(5):e20200226. doi: 10.36416/1806-3756/e20200226. eCollection 2020).

Answer: Thank you for this comment. We have revised it and marked in blue in page 3, line 44 .

3.Chest CT protocol: Authors could better explain the mean radiation dose exposure during CT.

Answer: Thank you for this comment. The mean radiation dose during CT is 7 mGy. We have revised it and marked in blue in page 6, line 89-90.

4.Chest CT image analysis: Authors could better clarify the years of experience on chest CT images.

Answer: Thank you for this comment. The years of experience on chest CT images analysis included the distribution of the lesion, the location of the lesion, the number of lobes involved, the characteristics of the lesion and external involvement. We have revised it and marked in blue in page 6, line 97-99 .

5. “… striated shadow or mesh shadow…” Authors could explain the mean of these words.

Answer: Thank you for this comment. “striated shadow” should be “linear opacities”.“mesh shadow” should be” reticulation”. We have revised it and marked in blue in the Manuscript and Revised Manuscript with Track Changes. 

All of the revisions are shown here. We would like to express our great appreciation to you and reviewers for comments on our paper.

Looking forward to hearing from you.

Best regards,

Shi-Ming Chen

---

## [Decision Letter · Decision Letter 1]

11 Feb 2021

PONE-D-20-31950R1

The characteristics and evolution of pulmonary fibrosis in COVID-19 patients as assessed by AI-assisted chest HRCT

PLOS ONE

Dear Dr. Chen,

Thank you for submitting your manuscript to PLOS ONE. After careful consideration, we feel that it has merit but does not fully meet PLOS ONE’s publication criteria as it currently stands. Therefore, we invite you to submit a revised version of the manuscript that addresses the points raised during the review process.

We look forward to receiving your revised manuscript.

Kind regards,

Giordano Madeddu

Academic Editor

PLOS ONE

Reviewers' comments:

Reviewer's Responses to Questions

**Comments to the Author**

1. If the authors have adequately addressed your comments raised in a previous round of review and you feel that this manuscript is now acceptable for publication, you may indicate that here to bypass the “Comments to the Author” section, enter your conflict of interest statement in the “Confidential to Editor” section, and submit your "Accept" recommendation.

Reviewer #2: All comments have been addressed

Reviewer #3: All comments have been addressed

2. Is the manuscript technically sound, and do the data support the conclusions?

Reviewer #2: Yes

Reviewer #3: Yes

3. Has the statistical analysis been performed appropriately and rigorously? 

Reviewer #2: Yes

Reviewer #3: Yes

4. Have the authors made all data underlying the findings in their manuscript fully available?

Reviewer #2: Yes

Reviewer #3: Yes

5. Is the manuscript presented in an intelligible fashion and written in standard English?

Reviewer #2: Yes

Reviewer #3: Yes

6. Review Comments to the Author

Reviewer #2: The revised manuscript is well written, the topic is original and Clear. Authors responded correctly to the requested reviews

Reviewer #3: The paper was revised in the correct way, although no article is mentioned in support of important laboratory prognostic indices in particular on liver function (https://doi.org/10.1111/eci.13427).

7. PLOS authors have the option to publish the peer review history of their article (what does this mean?). If published, this will include your full peer review and any attached files.

Reviewer #2: **Yes: **Paola Crivelli

Reviewer #3: No

---

## [Author Response · Author response to Decision Letter 1]

16 Feb 2021

Feb 16, 2021

Dear Editors and Reviewers:

Thank you for your letter and for the reviewers’ comments concerning our manuscript entitled “The characteristics and evolution of pulmonary fibrosis in COVID-19 patients as assessed by AI-assisted chest HRCT”. (ID: PONE-D-20-31950).

These comments are very constructive and helpful for revising our paper. We have discussed the comments carefully and have made revisions which we hope will be met with approval. They are all marked in blue in the revised version. The main answers to the reviewer’s comments are as follow: 

1.The paper was revised in the correct way, although no article is mentioned in support of important laboratory prognostic indices in particular on liver function (https://doi.org/10.1111/eci.13427) 

Answer: Thank you for this comment. This is a good suggestion. 

It was reported that the AST/ALT ratio on admission was significantly associated with in-hospital mortality in COVID-19 patients[https://doi.org/10.1111/eci.13427]. In this work, we found that there were significant relationships between the levels of AST and pulmonary fibrosis. AST displays the highest activity in the liver and skeletal muscle but also occurs in several tissues, including lungs, heart muscle, kidneys, pancreas, brain, leucocyte and erythrocytes. AST is less specific for liver damage compared to ALT[https://doi.org/10.1111/eci.13427]. Therefore, COVID-19 patients with a significant increase in AST may also have large damage in other tissues, including the lungs, which may affect pulmonary fibrosis at discharge. Larger studies are required to confirm the capacity of this parameter to independently predict pulmonary damage and fibrosis in this group. We have revised it and added the article as a reference in the revised version. 

2.There are some spelling errors in page 2, line 11-13. It should be as follow: The IL-6 level in the acute stage and albumin level were independent risk factors for pulmonary fibrosis. We have revised it and marked in blue in the revised version. 

3.There are some spelling errors in page 12, line 214-218. It should be as follow: Patients with or without pulmonary fibrosis had statistically significant differences in age, IL-6 levels, lymphocyte %, aspartate transaminase (AST), albumin, CRP/albumin ratio, platelet/lymphocyte ratio and some other indexes (Table 3), suggesting that these abnormal clinical indicators may be related to the pulmonary fibrosis. We have revised it and marked in blue in the revised version.

4.In addition, there are some errors in table 3 and table 4 because of the translation. We have revised it and marked in blue in the revised version.

All of the revisions are shown here. We would like to express our great appreciation to you and reviewers for comments on our paper.

Looking forward to hearing from you.

Best regards,

Shi-Ming Chen

---

## [Decision Letter · Decision Letter 2]

9 Mar 2021

The characteristics and evolution of pulmonary fibrosis in COVID-19 patients as assessed by AI-assisted chest HRCT

PONE-D-20-31950R2

Dear Dr. Chen,

We’re pleased to inform you that your manuscript has been judged scientifically suitable for publication and will be formally accepted for publication once it meets all outstanding technical requirements.

Kind regards,

Giordano Madeddu

Academic Editor

PLOS ONE

Additional Editor Comments (optional):

Reviewers' comments:

Reviewer's Responses to Questions

**Comments to the Author**

1. If the authors have adequately addressed your comments raised in a previous round of review and you feel that this manuscript is now acceptable for publication, you may indicate that here to bypass the “Comments to the Author” section, enter your conflict of interest statement in the “Confidential to Editor” section, and submit your "Accept" recommendation.

Reviewer #2: All comments have been addressed

Reviewer #3: All comments have been addressed

2. Is the manuscript technically sound, and do the data support the conclusions?

Reviewer #2: Yes

Reviewer #3: Yes

3. Has the statistical analysis been performed appropriately and rigorously? 

Reviewer #2: Yes

Reviewer #3: Yes

4. Have the authors made all data underlying the findings in their manuscript fully available?

Reviewer #2: Yes

Reviewer #3: Yes

5. Is the manuscript presented in an intelligible fashion and written in standard English?

Reviewer #2: Yes

Reviewer #3: Yes

6. Review Comments to the Author

Reviewer #2: All comments bave been addressed. The paper is well written, the topic is original and the aim i clear. For the anice-mentionned reatina, The paper can ne published. I suggest pubblication of this paper.

Reviewer #3: (No Response)

7. PLOS authors have the option to publish the peer review history of their article (what does this mean?). If published, this will include your full peer review and any attached files.

Reviewer #2: **Yes: **Paola Crivelli

Reviewer #3: No

---

## [Editor Report · Acceptance letter]

15 Mar 2021

PONE-D-20-31950R2 

The characteristics and evolution of pulmonary fibrosis in COVID-19 patients as assessed by AI-assisted chest HRCT 

Dear Dr. Chen:

I'm pleased to inform you that your manuscript has been deemed suitable for publication in PLOS ONE. Congratulations! Your manuscript is now with our production department. 

Kind regards, 

on behalf of

Dr. Giordano Madeddu 

Academic Editor

PLOS ONE